# Detection and genetic characterization of alphacoronaviruses in co-roosting bat species, southeastern Kenya

Joseph G. Ogola[1]*, Hussein Alburkat[2,3], Teemu Smura[2], Lauri Kareinen[2], Ravi Kant[2,4,5], Essi M. Korhonen[2,4], Tamika J. Lunn[6], Moses Masika[1], Paul W. Webala[7], Philip Nyaga[8], Omu Anzala[1], Olli Vapalahti[2,4,9], Kristian M. Forbes[10], Tarja A. Sironen[2,4]

1 Department of Medical Microbiology and KAVI Institute of Clinical Research, University of Nairobi, Nairobi, Kenya, 2 Departments of Virology, University of Helsinki, Helsinki, Finland, 3 Department of Microbiology, Faculty of Medicine Al-Muthanna University, Samawah, Iraq, 4 Department of Veterinary Biosciences, University of Helsinki, Helsinki, Finland, 5 Department of Tropical Parasitology, Institute of Maritime and Tropical Medicine, Medical University of Gdansk, Gdynia, Poland, 6 Odum School of Ecology, University of Georgia, Athens, Georgia, United States of America, 7 Department of Forestry and Wildlife Management, Maasai Mara University, Narok, Kenya, 8 Department of Veterinary Pathology, Microbiology and Parasitology, University of Nairobi, Nairobi, Kenya, 9 HUS Diagnostic Center, Virology and Immunology, Helsinki University Hospital, Helsinki, Finland 10 Department of Biological Sciences, University of Arkansas, Fayetteville, Arkansas, United States of America

* ogolajoseph2000@gmail.com

## Abstract

Bats are associated with some of the most significant and virulent emerging zoonoses globally, yet research and surveillance of bat pathogens remains limited across parts of the world. We surveyed the prevalence and genetic diversity of coronaviruses from bats in Taita Hills, southeastern Kenya, as part of ongoing surveillance efforts in this remote part of eastern Africa. We collected fecal and intestinal samples in May 2018 and March 2019 from 16 bat species. We detected one genus of coronavirus (alphacoronavirus), with an overall RNA prevalence of 6.5% (30/463). The prevalence of coronavirus RNA was 3.8% (9/235) and 11.6% (21/181) for the two most captured free-tailed bat species, *Mops condylurus* and *M. pumilus* respectively, with no detections from other bat species (0/90). Phylogenetic analyses based on the partial RNA-dependent RNA polymerase gene and whole genome sequences revealed that the sequences clustered together and were closely related to alphacoronavirus detected in free tailed bats in Eswatini, Nigeria and *Rhinolophus simulator bats* in South Africa. The sequences were more distantly related to alphacoronavirus isolated from *Chaerophon plicatus* bat species in Yunnan province, China and *Ozimops* species from southwestern Australia. These findings highlight coronavirus transmission among bats that share habitats with humans and livestock, posing a potential risk of exposure. Future research should investigate whether coronaviruses detected in these bats have the potential to spillover to other hosts.

**Data availability statement:** Sequences data from GeneBank are available for sharing, accession numbers can be found in Table 4. The raw NGS reads were deposited in the European Nucleotide Archive (ENA) at EMBL-EBI under accession number PRJEB96286.

**Funding:** This research was supported by the Jenny and Antti Wihuri Foundation (grant no. 358323, TAS), the Academy of Finland (grant no. 318726, OV), the Finnish Cultural Foundation (OV), the Jane and Aatos Erkko Foundation (OV and TAS), Helsinki University Hospital Funds (OV), Maj and Tor Nessling Foundation (EMK and JGO) and the Arkansas Biosciences Institute (KMF and TJL). The funders had no role in study design, data collection and analysis, decision to publish, or preparation of the manuscript.

**Competing interests:** The authors have declared that no competing interests exist.

## Author summary

Bats are known to carry several zoonotic pathogens with potential to cause serious illnesses and death in humans. Yet, surveillance on the pathogens they carry remains limited in much of the world. We studied the prevalence and diversity of coronaviruses from bats in Taita Hills, southeastern Kenya to better understand the circulation of these viruses and inform disease preparedness. We detected alphacoronaviruses in urban *Mops condylurus* and *M. pumilus* bat species. The bat alpha coronaviruses we detected were closely related to alphacoronaviruses that have been previously detected in bats elsewhere in Africa and distantly related to alphacoronavirus detected from *Chaerophon plicatus* bat species in Yunnan province, China and *Ozimops* species from southwestern Australia. This work demonstrates coronavirus circulation among bats that share habitats with people and livestock providing conditions that can lead to spillover. Identifying whether coronaviruses detected in these bats have the potential to infect other hosts is critical for developing countermeasures and mitigating potential outbreaks.

## 1 Introduction

Coronaviruses (CoVs) infect a wide range of hosts and are distributed throughout the world [1]. They can cause a number of diseases in humans, with variable severity ranging from asymptomatic to severe respiratory, gastrointestinal, liver, and neurologic diseases [2]. Coronaviruses have the potential to cause localized epidemics and global pandemics, as witnessed by three recent outbreaks: Severe Acute Respiratory Syndrome coronavirus (SARS-CoV), which emerged in a market in Guangdong province, China [3,4], Middle east respiratory syndrome (MERS) caused by MERS-CoV in the Arabian Peninsula [5,6], and the most recent global pandemic (COVID-19) caused by Severe Acute Respiratory Syndrome coronavirus 2 (SARS-CoV-2) [7]. The subfamily *Coronavirinae* is classified into four genera. Two of these, alpha- and beta-CoVs, can infect and sometimes cause disease in mammals, including humans. The remaining genera, gamma- and delta-CoVs, are mainly associated with birds [8,9].

All CoVs found in humans are likely zoonotic in origin, including those that cause mild but common respiratory diseases such as HCoV-HKU1 and HCoV-OC43 [10,11]. CoVs have a large genomic size ranging between 27 and 32 kb. As RNA viruses, they replicate by use of virus-encoded RNA polymerases, that are prone to high mutation rates. Phylogenetic studies indicate that cross-species transmission has occurred frequently during coronavirus evolution [12], facilitated by their high mutation rate, genetic recombination and ecological/anthropogenic factors. The occurrence of zoonotic spillover events may be linked to these viral features, together with the global distribution of CoVs in wildlife, giving CoVs opportunities to adapt to novel hosts [12,13].

Due to their high propensity for cross-species transmission and potential for disease emergence in humans, CoVs have become a focus for wildlife monitoring. Bats

(Mammalia: Chiroptera), in particular, have played a major role as the gene source for the evolution of alpha-CoVs and beta-CoVs (8) and have come to the forefront of coronavirus surveillance efforts. More than 4,000 coronavirus sequences from 14 bat families have been identified so far [14], yet the true diversity of bat coronaviruses is probably much higher. Bats are present in all six human-inhabited continents [15], though their species richness is highest in areas close to the equator [16]. This has correlated with the emergence and re-emergence of bat-borne viral pathogens in countries near the equator, including identified disease hotspots in Africa, the Americas, and Asia [17].

Hotspots along the equator are often also areas of high anthropogenic activities increasing contact between bats, livestock and humans, and escalating the risk of pathogen spillover. Bats continue to move closer to human settlements and sometimes live in the same houses used by people or abandoned buildings, further increasing the potential for spillover of the pathogens they host [18–20]. For example, serological evidence of human exposure to bat CoVs in rural China showed that spillover from bats might occur relatively commonly [21,22]. There is currently minimal surveillance of CoVs in eastern Africa despite Kenya being known for high diversity of bat fauna globally [23] and the risk CoVs pose to people. As part of ongoing wildlife surveillance for preparedness and prevention of zoonotic disease emergence, the purpose of this study was to survey the diversity and prevalence of CoVs from bats in Taita Hills, southeastern Kenya, a biodiversity hotspot, and characterize the genomes of identified viruses.

## 2 Methods

### 2.1 Ethics statement

Bat trapping and sample collections were conducted under permits from the Kenyan National Commission for Science, Technology and Innovation (permit no. NACOSTI/P/18/76501/22243) and the Kenya Wildlife Service (permit no. KWS/BRM/500), University of Nairobi Faculty of Veterinary Medicine; Biosafety, Animal use and Ethics committee (REF: FVM BAUEC/2018/180) and University of Arkansas Institutional Review Board (protocol #22012). Sample import to Finland was approved by the Finnish Food Safety Authority (EVIRA; 4250/0460/2016 and 2809/0460/2018).

### 2.2 Study area and sampling procedures

The study was conducted in the Taita Hills area, Taita Taveta County, southeast Kenya (Fig 1), as part of an ongoing virus surveillance project. Bats were captured in May 2018 and March 2019 using hand nets, and single- and triple-high mist nets at roosts and natural flyways. Captured bats were identified to species in the field using existing keys for bats of Kenya [23]. Non-conservation priority bat species (classified as least concern by the IUCN) were placed in clean individual cloth bags and retained for processing. For processing, bats were euthanized using an overdose of 4–5% isoflurane gas and immediately dissected for the collection of intestinal samples. Before terminal samples were collected, bat demographic information (sex, age and reproductive status) was recorded alongside standard measurements (weight and forearm length). Bats were considered reproductive if female bats were gestating or lactating at the time of capture. Gestation was observed by gentle palpation of the abdomen for the presence of uterine bulge while lactating bats were identified by the presence of engorged nipples either with or without bare patches of skin [28]. Male bats were considered reproductive if they were scrotal and not juvenile. Bats were classified as non-reproductive if there was no detectable pregnancy/lactation or scrotum among female and males respectively. Fecal samples were collected from the individual cloth bags when available. Both intestinal tissues and fecal samples were placed into separately marked tubes with RNAlater (Qiagen, Hilden, Germany), stored at -20°C, and later shipped on dry ice to Helsinki, Finland for laboratory testing.

### 2.3 Sample processing and detection of coronaviruses

Bat fecal and intestinal samples were treated with Tripure (Roche, http://www.roche.com), according to the manufacturer's instructions, to inactivate any potentially biohazardous agents before RNA extractions. Extracted RNAs were eluted in 50 µL of RNase-free water, aliquoted into 15 µL, quantified by NanoDrop spectrophotometer (Thermo Fisher Scientific),

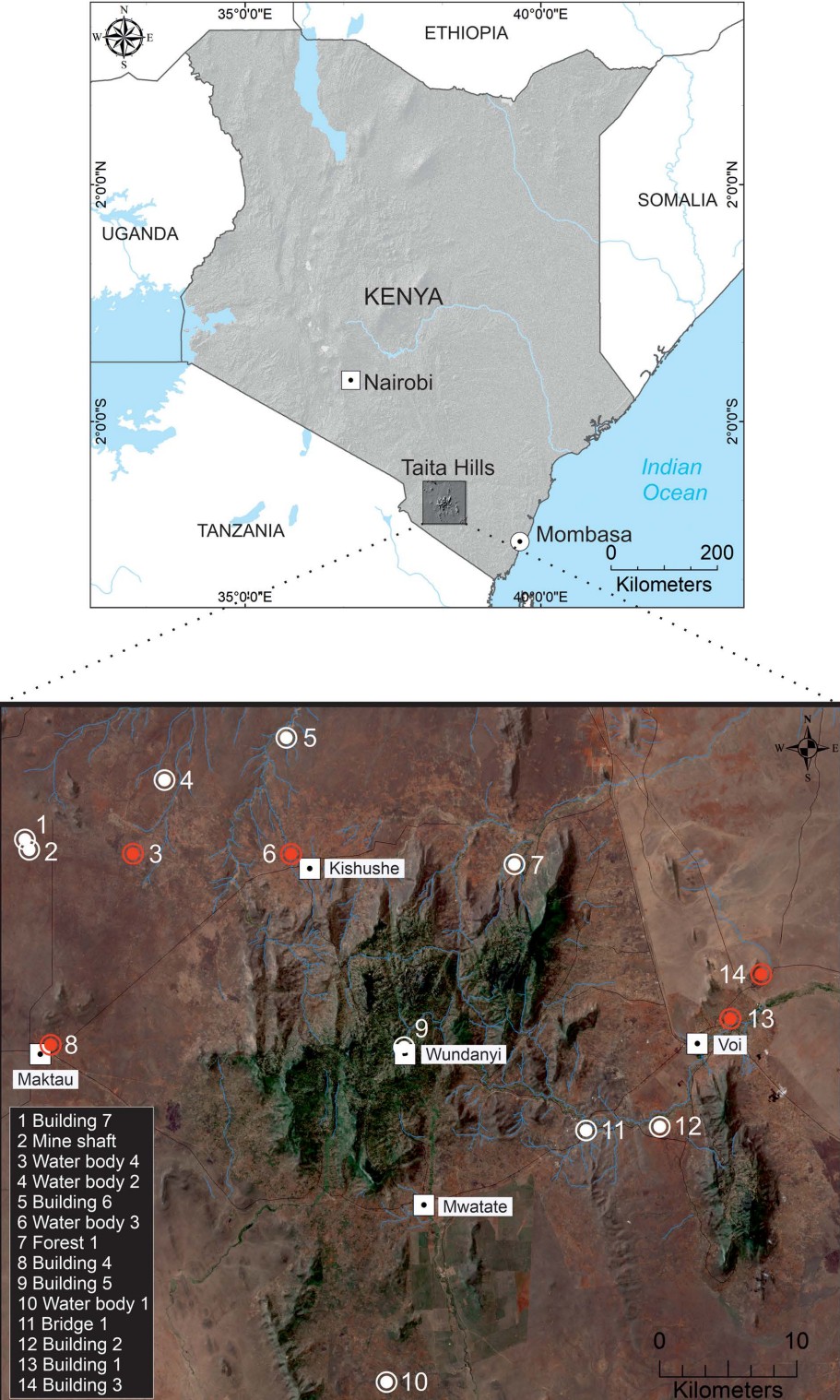

**Fig 1. Map of Taita Hills showing the spatial location of sites where bat species were captured in May 2018 and March 2019.** The white large dots and red dots indicate the bat trapping sites and positive CoV sites respectively. The map was created using ArcMap (version 10.8). Shape files for country boundaries, rivers, and lakes, as well as the 30-meter resolution SRTM Kenya Digital Elevation Model (DEM), were obtained from various open

data sources [24–26]. True color background data of Taita Hills was derived from Sentinel-2 MSI Level-1C imagery [27] (ESA/Copernicus, tile T37MCR, 09 September 2025).

and stored at –80°C for downstream analysis. Samples were then screened with qScript One-Step SYBR Green qRT-PCR Kit, Low ROX (Quanta Biosciences, Beverly, MA) using primers targeting the CoV RdRP-gene (11-FW 5′ TGAT-GATGSNGTTGTNTGYTAYAA 3′ and 13-RV GCATWGTRTGYTGNGARCARAATTC 3′ which are able to detect all four genera of coronaviruses currently known [29]). Positive samples were transcribed to cDNA using SuperScript IV One-Step RT-PCR Kit (ThermoFisher Scientific, Invitrogen, USA) with only the first reaction of the nested RT-PCR protocol used [30] and PCR amplicons were sequenced using Sanger sequencing.

A total of 12 positive coronavirus RNA samples from the two species were further subjected to next-generation sequencing (NGS). For this, the CoV positive RNA samples were reverse transcribed and amplified using the WTA2 kit (Sigma). Products were then purified based on the manufacturer's instructions with the GeneJET PCR purification kit (Thermo Scientific). NGS libraries were prepared with the Nextera XT DNA Sample Preparation and the Nextera XT Index Kit for 24 Indexes (Illumina), and sequencing was performed with MiSeq (Illumina), using the MiSeq Reagent Kit v2-150.

Since NGS resulted in fragmentary CoV genomes, specific primer pairs were designed in the genomic regions flanking gaps and these regions were amplified with PCR using 10.0µl of 2x Phusion Flash PCR Master Mix (Thermo Scientific), 2.5µl of 10µM primers and 2.5µl of water. The cycling conditions were as follows: initial denaturation at 9ºC for 10s, followed by 30 cycles of the following steps of denaturation at 98ºC for 1s, annealing at 56ºC for 5s and extension at 72ºC for 7s, and final extension at 72ºC for 1 min. PCR products were run on a 1.5% agarose gel, purified as described earlier, and sequenced with Sanger sequencing using PCR primers. If the gap-filling PCR was unsuccessful, Sanger sequencing was performed with the original complete genome PCR product as a template.

## 2.4 Statistical analysis

Field and laboratory data were stored in Microsoft Excel files. They were imported into SPSS software for descriptive and statistical analysis. Dependent variables that were analyzed included bat species, age, sex and reproductive status and location of bat trapping. We explored the association between the prevalence of CoV RNA and the dependent variables using univariate logistic regression.

## 2.5 Sequence analysis

The raw sequence reads were quality filtered, trimmed, de-novo assembled, and annotated using fastp v.1.0.1 [31], MEGAHIT v.1.2.8 [32] and SANSparallel [33], respectively, implemented in Lazypipe pipeline [34]. Thereafter, for the samples in which alphacoronavirus sequences were detected, the quality filtered reads (quality score threshold of 30, maximum allowed percentage of low-quality bases 40, minimum read length: 25, sliding window size: 20 mean with a mean quality threshold of 30; and base correction for paired-end reads enabled) were remapped against the near-complete genome sequence from sample X167 using BWA-MEM [35] implemented in HAVoC pipeline [36]. Thereafter, sequence reads were filtered by mapping quality (MAPQ) using a threshold of 30, followed by consensus calling with SAMtools [37] and BCFtools [38]. A summary of sequencing data quality and alignment metrics is provided in S1 Table.

For phylogenetic analysis, all available complete or nearly complete alphacoronavirus genomes were downloaded from NCBI GenBank and aligned using MAFFT [39]. The alignment was further subsampled to include only one representative of sequence groups with less than 5% pairwise amino acid sequence divergence. The complete genome alignment was divided into 5 alignments, representing each gene (coding for ORF1ab, spike, NS3, envelope and membrane proteins) in the alphacoronavirus core genome (i.e., genes present in all the members of the

alphacoronavirus genus). Phylogenetic trees were constructed using maximum likelihood (ML) method implemented in IQ-TREE2 v.2 [40], employing ModelFinder [41] algorithm to determine the optimal protein substitution model, and the ultrafast bootstrap UFBoot2 [42] algorithm to compute 1000 bootstrap pseudo replicates. The final trees were visualized with iTOL v5 [43].

## 3 Results

A total of 510 bats were captured during the study period, comprising mainly *Mops condylurus* (*n* = 237) and *M. pumilus* (*n* = 183), which were the focus species of our field trapping due to their close association with humans as species that commonly roost in anthropogenic buildings. In addition, 90 other bats from 14 species were captured and screened (Table 1). The overall prevalence for CoV RNA was 6.5% (30/463) [95% CI 4.4-9.1%] and varied markedly among sites (Table 2), sometimes skewed by a small number of captured bats.

**Table 1. Species of bats captured, number sampled and the prevalence of CoV RNA based on bat sex and reproductive status. Bats were classified as reproductive if they were reproductively active at the time of capture (females: pregnant or lactating; males: scrotal) or as non-reproductive if they were not.**

| Species | Males | Females | Coronavirus prevalence | |
| --- | --- | --- | --- | --- |
| | | | Reproductive | Non-Reproductive |
| *Mops condylurus* | 6/102 (5.9%) | 3/133 (2.3%) | 5/96 (5.2%) | 4/139 (2.9%) |
| *Mops pumilus* | 10/92 (10.9%) | 11/89 (12.4%) | 2/27 (7.4%) | 19/154 (12.4%) |
| *Cardioderma cor* | – | – | 0/12 (0.0%) | – |
| *Epomophorus wahlbergi* | – | – | 0/3 (0.0%) | 0/3 (0.0%) |
| *Rousettus aegyptiacus* | – | – | – | 0/4 (0.0%) |
| *Scotophilus andrewreborii* | – | – | 0/4 (0.0%) | 0/2 (0.0%) |
| *Scotoecus cf. hirundo* | – | – | 0/2 (0.0%) | 0/3 (0.0%) |
| *Neoromicia spp* | – | – | 0/1 (0.0%) | 0/1 (0.0%) |
| *Rhinolophus hildebrandtii* | – | – | 0/1 (0.0%) | 0/1 (0.0%) |
| *Rhinolophus webalai* | – | – | 0/1 (0.0%) | 0/1 (0.0%) |
| *Rhinoliphus clivosus* | – | – | – | 0/2 (0.0%) |
| *Hipposideros caffer* | – | – | – | 0/1 (0.0%) |
| *Nycticeinops schlieffeni* | – | – | 0/1 (0.0%) | 0/1 (0.0%) |
| *Nycteris thebaica* | – | – | – | 0/1 (0.0%) |
| *Stenonycteris lanosus* | – | – | 0/1 (0.0%) | – |
| *Lavia frons* | – | – | 0/1 (0.0%) | – |
| **Total** | – | – | **7/150 (4.7%)** | **23/313 (7.4%)** |

**Table 2. The prevalence of coronavirus RNA in *Mops condylurus* and *M. pumilus* trapped from different sites at Taita hills, Kenya.**

| Bat trapping site | Habitat | *Mops condylurus* | *Mops pumilus* |
| --- | --- | --- | --- |
| Building 2 | Human dwelling | – | 0/28 (0.0%) |
| Building 4 | Commercial building | 2/21 (9.5%) | 3/110 (2.7%) |
| Bridge 1 | Water bridge | 1/110 (0.9%) | 0/31 (0.0%) |
| Forest 1 | Forest | 0/1 (0.0%) | – |
| Water body 3 | Riverbank | 5/13 (38.5%) | 0/3 (0.0%) |
| Water body 4 | Riverbank | 1/1 (100%) | 0/2 (0.0%) |
| Building 3 | Human dwelling | – | 16/88 (18.2%) |
| Building 1 | Human dwelling | – | 2/8 (25.0%) |

The two species, *Mops condylurus* and *Mops pumilus*, had significantly different CoV prevalence: prevalence was lower in *M. condylurus* bats 3.8% (9/235) [95% CI 1.8-7.2%] as compared to *M. pumilus* bats 11.6% (21/181) [95% CI 7.3-17.2%], with odds ratio of 0.303 (0.135 -0.680), and p-value of 0.004 (Table 3). CoV prevalence varied slightly also based on reproductive status; however, the difference was not significant.

### 3.1 Phylogenetic analysis of partial RdRp coding sequences

Thirty partial RdRp nucleotide sequences were obtained from coronavirus screening PCR amplicons. The initial annotation using blastn algorithm indicated that they had 96–99% nucleotide identity against alphacoronaviruses sequenced from genus *Chaerophon* bats from Kenya, Eswatini, and Nigeria [44–46] as well as *Rhinolophus simulator* bats from South Africa [47]. Three short sequences were excluded from further analysis due to the short length of nucleotide bases and the presence of ambiguous bases and the remaining sequences were submitted to the NCBI Genbank with accession numbers (Table 4). The raw NGS reads were deposited in the European Nucleotide Archive (ENA) at EMBL-EBI under accession number PRJEB96286. In the phylogenetic analysis, these sequences clustered together and the sequences from Kenya interspersed with those from the other African countries (Fig 2). Notably, all the sequences from each sampling location do not cluster together, but form distinct clusters interspersed throughout the tree. On the other hand, some sequences sampled from two different locations were identical to each other. For example, three sequences from Maktau and 10 sequences from Voi (Fig 2) were identical, although Maktau and Voi are located 51.7 km apart. Further, the cluster containing the viruses sequenced in this study was more distantly related to alphacoronaviruses sequenced from *Otomops harrisoni*, Mount Suswa, Kajiado county, Kenya [48], *Otomops martienssini,* Rwanda, *Chaerophon* sp. bat from Kenya [45] as well as those sequenced from *Hipposideros armiger* and *Hipposideros pomona* bats in China [49].

### 3.2 Genomic characterization of complete coding sequences

Two nearly complete alphacoronavirus genomes were obtained using NGS. Of these, complete coding regions were obtained from the strain X167, whereas strain X152 contained two gaps (amino acids 1–42 and 71–114), which we were not able to close with specific PCRs due to the lack of sample availability. The genome organization of these strains was canonical to subgenus *Decacovirus* (Fig 3), where ORF1ab with (ribosomal slippage site) is followed by spike (S), non-structural protein (NS3), envelope protein (E), matrix protein (M) and nucleoprotein (N) coding regions, as well as ORFx. The spike protein contains the S1/S2 cleavage region, but they do not contain a polybasic cleavage site. Furthermore, Hidden Markov Model search suggested that the coded proteins contained conserved functional domains typical for alphacoronaviruses (S2 Table).

Table 3. Univariate logistic regression for the presence of bat coronaviruses and the species, sex and reproductive status of captured *Mops condylurus* and *M. pumilus* bats.

| Parameter | | Coronavirus prevalence (n/N (%)) | OR (95%) | P value |
|---|---|---|---|---|
| Species | *Mops pumilus* | 21/181 (11.6%) | Reference | |
| | *Mops condylurus* | 9/235 (3.8%) | 0.303 (0.135 -0.680) | 0.004 |
| Sex | Male | 16/217 (7.4%) | Reference | |
| | Female | 14/246 (5.7%) | 0.758 (0.361 -1.592) | 0.464 |
| Reproductive status | Reproductive | 7/150 (4.7%) | Reference | |
| | Non-reproductive | 23/313 (7.3%) | 1.620 (0.679 -3.865) | 0.277 |

**Table 4. Complete coding sequences and partial RdRp sequences from *M. condylurus* and *M. pumilus* submitted to the NCBI GenBank with accession numbers.**

| No. | Bat ID | Species | Genome | Genbank Accession |
|---|---|---|---|---|
| 1 | X193 | M. pumilus | Complete CDS | PV101458.1 |
| 2 | X152 | M. pumilus | Partial CDS* | PV101459.1 |
| 3 | X162 | M. pumilus | Complete CDS | PV101460.1 |
| 4 | X167 | M. pumilus | Complete CDS | PV101461.1 |
| 5 | X152 | M. pumilus | Partial RdRp | PV054882.1 |
| 6 | X153 | M. pumilus | Partial RdRp | PV054883.1 |
| 7 | X154 | M. pumilus | Partial RdRp | PV054884.1 |
| 8 | X162 | M. pumilus | Partial RdRp | PV054885.1 |
| 9 | X193 | M. pumilus | Partial RdRp | PV054886.1 |
| 10 | X198 | M. pumilus | Partial RdRp | PV054887.1 |
| 11 | X201 | M. pumilus | Partial RdRp | PV054888.1 |
| 12 | X207 | M. pumilus | Partial RdRp | PV054889.1 |
| 12 | X226 | M. pumilus | Partial RdRp | PV054890.1 |
| 14 | X244 | M. condylurus | Partial RdRp | PV054891.1 |
| 15 | X329 | M. condylurus | Partial RdRp | PV054892.1 |
| 16 | X343 | M. pumilus | Partial RdRp | PV054893.1 |
| 17 | X348 | M. pumilus | Partial RdRp | PV054894.1 |
| 18 | X388 | M. condylurus | Partial RdRp | PV054895.1 |
| 19 | X173 | M. pumilus | Partial RdRp | PV054898.1 |
| 20 | B241 | M. condylurus | Partial RdRp | PV054899.1 |
| 21 | X195 | M. pumilus | Partial RdRp | PV054900.1 |
| 22 | X209 | M. pumilus | Partial RdRp | PV054901.1 |
| 23 | X213 | M. pumilus | Partial RdRp | PV054902.1 |
| 24 | X216 | M. pumilus | Partial RdRp | PV054903.1 |
| 25 | X154 | M. pumilus | Partial RdRp | PV054904.1 |
| 26 | X177 | M. pumilus | Partial RdRp | PV054905.1 |
| 27 | X344 | M. pumilus | Partial RdRp | PV054906.1 |
| 28 | X04 | M. pumilus | Partial RdRp | PV054907.1 |
| 29 | X06 | M. pumilus | Partial RdRp | PV054908.1 |
| 30 | B207 | M. condylurus | Partial RdRp | PV054909.1 |
| 31 | B212 | M. condylurus | Partial RdRp | PV054910.1 |
| 32 | B223 | M. condylurus | Partial RdRp | PV054911.1 |

* Missing amino acids 1–42 and 71–114 in ORF1a.

### 3.3 Phylogenetic analysis based on complete coding regions

Phylogenies based on the amino acid sequences of core genes (ORF1ab, spike, NS3, E, M, N) indicated that our sequences clustered together on the basis of all analyzed proteins (Fig 4). The closest relatives for our sequences were alphacoronaviruses sequenced from *Chaerephon pumilus* fecal samples from Eswatini, southern Africa [44] and oral/rectal swabs of *Mops condylurus* bats from Nigeria, western Africa [46] (Fig 4). These African bat-associated viruses formed a larger cluster with more distantly related alphacoronaviruses Yunnan/CpYN11/2019 and WA3607 (subgenus *Decacovirus*) sequenced from *Chaerephon plicatus* bat fecal sample from Yunnan province, China [50] and *Ozimops* sp from southwestern Australia [51], respectively (S1A-E Fig). Notably, while these sequences formed

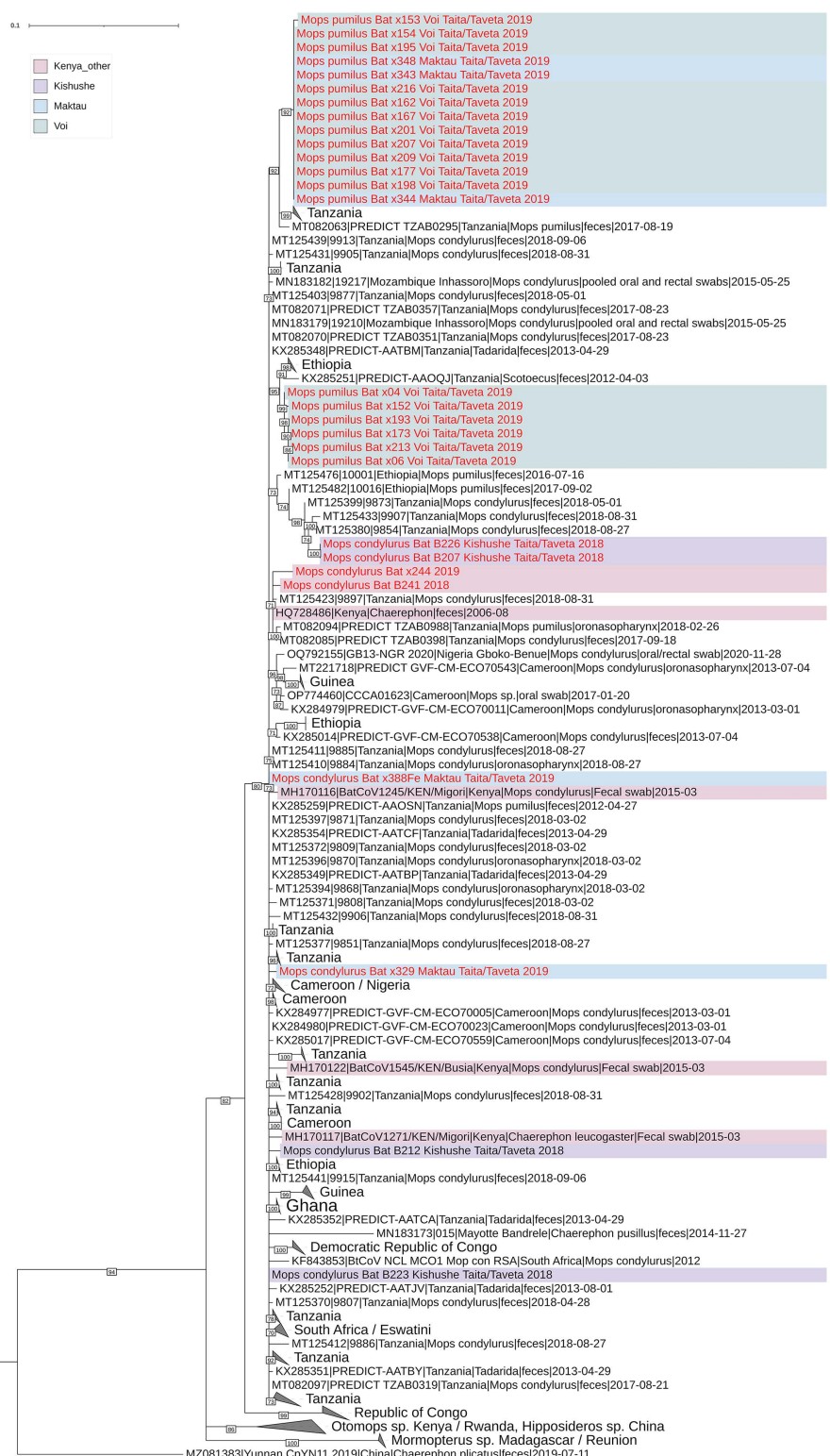

**Fig 2. Phylogenetic tree based on a 378 nucleotides long alignment of partial RdRp sequences encoding the conserved polymerase motif B.**
The tree was constructed using maximum likelihood method implemented in IQTree2 software with TN+F+R3 substitution model and 1000 ultrafast bootstrap replicates. The tree was rooted to mid-point, the nodes with bootstrap support less than 70 were polytomized and the clades with sequences from only one country in eastern Africa, or 2-3 countries from western or southern Africa were collapsed for the sake of clarity.

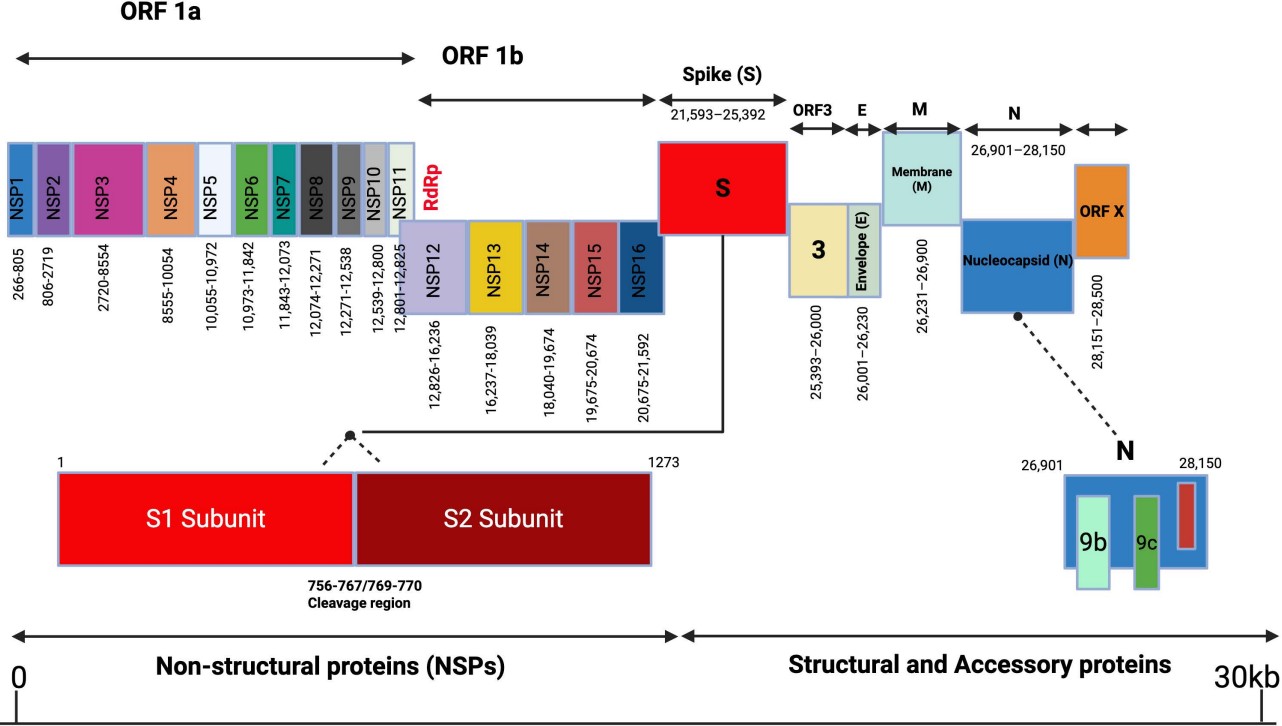

**Fig 3. Gene composition of alphacoronavirus.** This figure illustrates the genome organization of an alphacoronavirus, highlighting the nucleotide lengths of individual genes along the genome map. Arrows represent the relative positions and sizes of each gene. The figure was created using BioRender.

a monophyletic group based on spike amino acid sequences, three other alphacoronavirus sequences were also included in this cluster. Alphacoronavirus isolate Tb2 from *Tadarida brasiliensis* bat fecal sample from Santa Fe province, Argentina [52], grouped together with WA3607 strain, and the *Miniopterus* sp alphacoronavirus strains HKU8 from Hong Kong [53] and BtMf-AlphaCoV/HuB2013 from Hubei province, China [54] formed their own node within this cluster (S1A-E Fig).

To further study the evolutionary relationship between our sequences and the related sequences, we included partial alphacoronavirus sequences sampled from *Otomops harrisoni*, Mount Suswa, Kajiado county, Kenya (partial ORF1a and complete ORF1b available) [48] and *Chaerophon* sp. bat from Kenya (complete ORF1b and onwards available) [45] to the analysis (Fig 4). Based on both ORF1a and ORF1b regions, the sequences from *O. harrisoni* form a sister clade to our sequences and sequences from Eswatini and Nigeria. In ORF1a and spike regions, our sequences cluster together with those from Eswatini. This clustering pattern remains similar throughout NS3, E, M and N coding regions. However, based on ORF1b, our sequences, along with those from Eswatini, Nigeria and KY22 strain from Kenya each form their separate lineage with unresolved deeper phylogenetic relationships.

## 4 Discussion

Predicting future coronavirus outbreaks requires enhanced surveillance to investigate coronavirus prevalence and distribution among bats alongside characterizing their genetic features. We identified coronaviruses in two bat species in the Taita Hills of southeastern Kenya and characterized the virus genomes. This information helps fill an important gap in knowledge about coronavirus diversity in underexplored and high-risk parts of the world.

## ORF1a

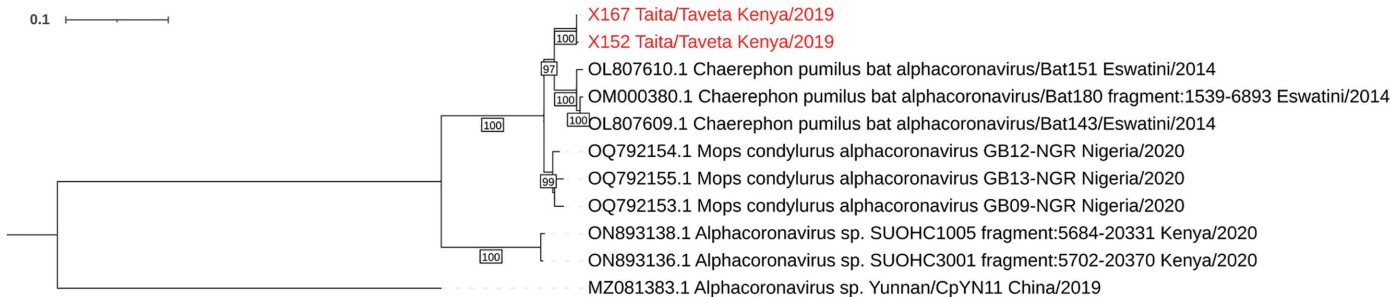

## ORF1b

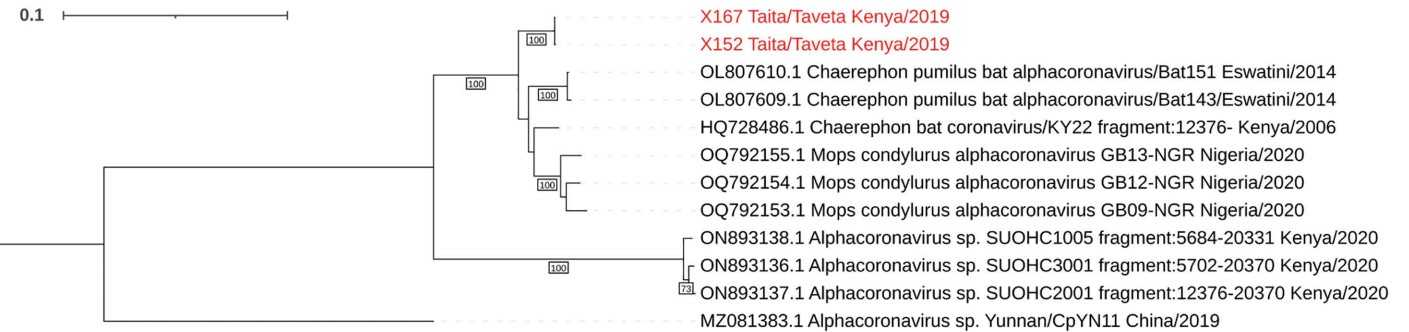

## SPIKE

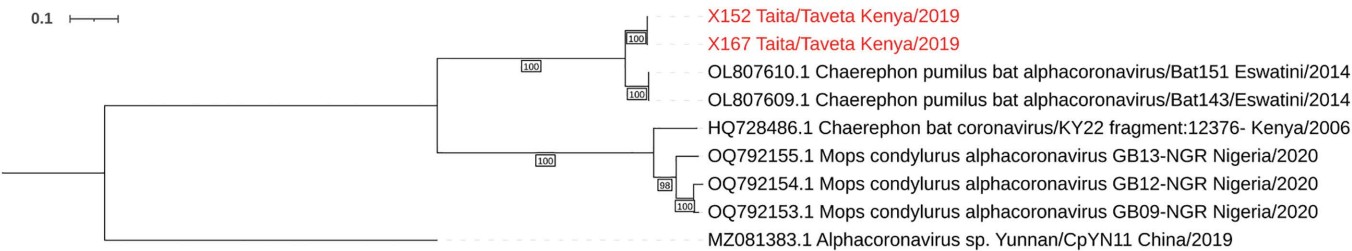

**Fig 4. Phylogenetic trees based on (a) ORF1a, (b) ORF1b and (c) spike coding regions of the complete genome sequences from this study, as well as closely related sequences, of which nearly complete genomes were available.** The trees were constructed with maximum likelihood method implemented in IQTree2 software with best fit models GTR+F+I+G4 for ORF1a and ORF1b, and TIM3+F+R3 for spike coding region and 1000 ultrafast bootstrap replicates. The tree was rooted to mid-point.

The prevalence of CoVs (6.5%) in our study area was consistent with previous studies on coronavirus prevalence in bats. For example, Cappelle et al. reported coronavirus prevalence of 4.2% (24/573) in Cambodian bats from Kampot and 4.75% (22/463) in flying foxes from Kandal [55]. Studies by Xu et al. reported coronavirus in 5.3% (50/951) of bats in the Tibet autonomous Region of China [56]. Notably, coronavirus prevalence varies greatly between studies, with some of the highest coronavirus prevalence's reported by Ge et al. 50% (138/276) in China [57], Balboni et al. 42% (19/45) in Italy [58] and by Tsuda et al. 29.6% (53/179) in the Philippines [59].

CoV prevalence in our study varied across the species and the capture sites, with prevalence being significantly higher in *M. pumilus* than in *M. condylurus*. The variability in host prevalence potentially highlights the complexity of exposure, infection, and immune dynamics over space and time. Although we only detected alphacoronaviruses in *M. pumilus* and *M. condylurus*, alphacoronaviruses and other novel coronaviruses have previously been found in some other bats in Kenya [48]. For example, Tong et al (2006) detected coronaviruses in different species of bats including *Miniopterus inflatus*, *Rousettus aegytiacus* and *Eidolon helvum* and others [60]. In addition, coronaviruses have also been previously detected in *Eidolon helvum* urban roost in Tanzania, frugivorous bats in Madagascar and different species of bats from Mozambique and Madagascar [61–63].

We detected coronaviruses in bats trapped from houses occupied by humans and also water points which are used by both humans and livestock for drinking. This poses a significant zoonotic risk for the spillover of bat borne zoonotic viruses, particularly coronaviruses which have been documented by studies across Africa [64–65]. In most rural communities in Africa, molossid bats, in particular *M. pumilus* and *M. condylurus* often roost in human occupied buildings, including roofs, ceilings and wall cavities, increasing contact with humans and livestock. Livestock in particular may act as an intermediate or amplifying host. Such close interaction increases the likelihood of viral exposure through contamination with bat excreta, saliva, or urine. In addition to the low public awareness of zoonotic risks, the risk of exposure is amplified in rural communities due to limited resources and housing conditions that may be inadequate to prevent wildlife invasion.

The partial coronavirus RdRp sequences obtained in our study showed a clustering pattern with Kenyan sequences, interspersed among bat sequences from other African countries. We observed two large clusters of nearly identical sequences; one cluster included bats trapped from two different colonies in two different locations (Voi and Maktau, 51.7 kms apart) while the second cluster included bats roosting in the same location (Voi, 3.5 kms apart). While *Mops condylurus and Mops pumilus* bats are thought to travel short distances due to their feeding behavior [66], detection of identical sequences in two different locations in Taita, as well as the intermixing of sequences from different countries in the phylogenetic tree, demonstrates likely longer distance movements of molossid bats.

In the past decade, knowledge on bat coronavirus ecology and epidemiology has significantly increased. Alphacoronaviruses have been reported in several bat populations and other mammalian hosts, with several lineages in bat species that often roost near human settlements and agricultural environments [18–19]. Severe acute diarrhea syndrome coronavirus (SADS-CoV), an emerging virus responsible for severe, acute diarrhea in piglets in China is one of the most recent bat-derived alphacoronavirus [67]. Studies by Antony et al. estimated approximately 3,204 bat coronaviruses worldwide [68]. Similar to most emerging zoonoses, coronavirus spillover and emergence may be linked to high mutation rates, potential for recombination and anthropogenic changes such as deforestation, agricultural intensification and urbanization [17]. Further studies on bat coronavirus could focus on extensive surveillance of coronavirus in different bat species and also animals with close bat contact for coronavirus spillover and emergence in rural communities. The recent COVID-19 pandemic underscores the need for an increased assessment of CoV diversity and spillover risk at local and regional levels.

## Supporting information

**S1 Fig.** A. Phylogenetic trees based on amino acid sequences of the representatives of all available alphacoronavirus species. The trees were constructed with maximum likelihood method implemented in IQTree2 software with best fit substitution models LG + F + R10, WAG + F + R7, LG + F + I + G4, LG + F + I + G4 and LG + F + R5 for ORF1ab

(A), spike (B), ORF3 (C), M (D) and N (E) respectively. The clusters are coloured on the basis of ORF1ab clustering pattern.
(PPTX)

**S1 Table. Summary of sequencing quality and alignment metrics.**
(DOCX)

**S2 Table. The results of Hidden Markov Model (HMMER v3.4) search against Pfam database.**
(DOCX)

## Acknowledgments

We wish to acknowledge the training and support from the University of Nairobi's Building Capacity for Writing Scientific Manuscripts (UANDISHI) Program at the Faculty of Health Sciences. The training was funded in part through the ADVANCE program at IAVI. This work is made possible by the support of the American People through the U.S. President's Emergency Plan for AIDS Relief (PEPFAR) through United States Agency for International Development (USAID). The contents of this study are the sole responsibility of the authors and do not necessarily reflect the views of PEPFAR, USAID, or the United States Government. We also thank Ruut Uusitalo for preparing the map image.

## Author contributions

**Conceptualization:** Joseph G. Ogola, Lauri Kareinen, Paul W. Webala, Philip Nyaga, Omu Anzala, Olli Vapalahti, Kristian M. Forbes, Tarja A. Sironen.

**Formal analysis:** Joseph G. Ogola, Hussein Alburkat, Teemu Smura, Ravi Kant, Tarja A. Sironen.

**Investigation:** Joseph G. Ogola, Hussein Alburkat, Teemu Smura, Lauri Kareinen, Paul W. Webala, Philip Nyaga, Kristian M. Forbes, Tarja A. Sironen.

**Writing – review & editing:** Joseph G. Ogola, Hussein Alburkat, Teemu Smura, Lauri Kareinen, Ravi Kant, Essi M. Korhonen, Tamika J. Lunn, Moses Masika, Paul W. Webala, Philip Nyaga, Omu Anzala, Olli Vapalahti, Kristian M. Forbes, Tarja A. Sironen.

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
