## [Decision Letter · Decision Letter 0]

10 Mar 2025

Detection and genetic characterization of alphacoronaviruses in co-roosting bat species, southeastern Kenya

Dear Dr. Ogola,

Thank you for submitting your manuscript to PLOS Neglected Tropical Diseases. After careful consideration, we feel that it has merit but does not fully meet PLOS Neglected Tropical Diseases's publication criteria as it currently stands. Therefore, we invite you to submit a revised version of the manuscript that addresses the points raised during the review process.

Please submit your revised manuscript within 60 days May 09 2025 11:59PM. If you will need more time than this to complete your revisions, please reply to this message or contact the journal office at plosntds@plos.org. Please include the following items when submitting your revised manuscript:

We look forward to receiving your revised manuscript.

Kind regards,

Michael W Gaunt, PhD

Academic Editor

Andrea Marzi

Section Editor

Shaden Kamhawi

co-Editor-in-Chief

Paul Brindley

co-Editor-in-Chief

**Additional Editor Comments:**

Comments from the AE

Please note that reviewer 2 has questioned the phylogenetics.

The authors originally stated:

"For recombination analysis, a subset of sequences, representing the virus strains that showed the closest homology to those sequenced in this study, were subjected to similarity plot and bootscanning analyses using SimPlot++ software [36] and RDP, GENECONV, BootScan, MaxChi, Chimaera, SiScan, and 3Seq [37–43], as implemented in RDP5 software [44]."

This statement is no longer accurate. It is important to note that phylogenetic incongruence in viruses is not necessarily indicative of recombination. For instance, Gaunt et al. (2022) in Virus Evolution demonstrated that in flaviviruses, strong phylogenetic incongruence can arise not from recombination but from extreme selective pressures. RNA viruses possess exceptional mutational capacities and can undergo reversion under selection pressure, independently generating similar mutations within contiguous loci.

Additionally, I am not convinced that the manuscript provides evidence of phylogenetic incongruence with the possible small exception of X162. The inference in Figure 2 relies on a bootstrap value of 73 - between Spike and the ORF1/ORF2, which is insufficient for robust conclusions, there would need to be alternative approaches including Bayes and that woudl need to return a robust value for the branch. Figure 3 does indicate some incongruence for X162, but this is not evident in Figure 2. I recommend a thorough reassessment of all phylogenetic incongruence analyses, as the observed signal appears to be a minor feature centered around X162. If the manuscript is resubmitted, I will conduct a full review of all supporting calculations. Also the way the ms is written, it overstating a minor incongruence in a single taxa X162 which only appears present in one analysis.

Moreover, Figure 3 is poorly presented, making it unclear which specific analyses the authors have performed. For instance, bootscanning analysis typically results in a plot, which is absent from the figure. Clarification is needed in this regard. I am not at all clear what the authors have done for Figure 3 and accurate descriptions of all analytics are required to ensure transparency.

It is important that the authors resolve the criticisms of their phylogenetic methods and do not overstate their findings because the standards of phylogenetics in PLOS NTD are much higher than the authors have presented.

The supplementary trees are also missing entirely, which must be rectified. To prevent formatting issues, I strongly recommend submitting these as a PDF.

Additionally, the table categorizing "Reproductive" and "Non-Reproductive" is confusing for readers unfamiliar with bat biology. Consider rewording or providing additional explanation to ensure clarity.

Finally, I urge the authors to carefully proofread the manuscript for grammatical errors. I can suspect I could produce an extensive list of 50 corrections and if so each would need to be addressed individually.

**Journal Requirements:**

At this stage, the following Authors/Authors require contributions: Joseph Ganda Ogola, Hussein Alburkat, Teemu Smura, Lauri Kareinen, Ravi Kant, Essi M. Korhonen, Tamika J. Lunn, Moses Masika, Paul W. Webala, Philip Nyaga, Omu Anzala, Olli Vapalahti, Kristian M. Forbes, and Tarja A. Sironen. Please ensure that the full contributions of each author are acknowledged in the "Add/Edit/Remove Authors" section of our submission form.

Potential Copyright Issues:

- Figure 1. Please provide a direct link to the base layer of the map (i.e., the country or region border shape) and ensure this is also included in the figure legend; and provide a link to the terms of use / license information for the base layer image or shapefile. We cannot publish proprietary or copyrighted maps (e.g. Google Maps, Mapquest) and the terms of use for your map base layer must be compatible with our CC BY 4.0 license.

5) When completing the data availability statement of the submission form, you indicated that you will make your data available on acceptance. We strongly recommend all authors decide on a data sharing plan before acceptance, as the process can be lengthy and hold up publication timelines. Please note that, though access restrictions are acceptable now, your entire data will need to be made freely accessible if your manuscript is accepted for publication. This policy applies to all data except where public deposition would breach compliance with the protocol approved by your research ethics board. If you are unable to adhere to our open data policy, please kindly revise your statement to explain your reasoning and we will seek the editor's input on an exemption. Please be assured that, once you have provided your new statement, the assessment of your exemption will not hold up the peer review process.

6) Please ensure that the funders and grant numbers match between the Financial Disclosure field and the Funding Information tab in your submission form. Note that the funders must be provided in the same order in both places as well.

**Reviewers' Comments:**

Reviewer's Responses to Questions

**Key Review Criteria Required for Acceptance?**

**Methods:**

-Are the objectives of the study clearly articulated with a clear testable hypothesis stated?

-Is the study design appropriate to address the stated objectives?

-Is the population clearly described and appropriate for the hypothesis being tested?

-Is the sample size sufficient to ensure adequate power to address the hypothesis being tested?

-Were correct statistical analysis used to support conclusions?

-Are there concerns about ethical or regulatory requirements being met?

Reviewer #1: Methods appear appropriate and well designed.

Some additional details on collection sites would be useful.

Reviewer #2: I have no ethical concerns related to data collection or sampling. Some statistical problems are addressed in the section about Results below.

Reviewer #3: The study design and methods used are appropriate and well explained.

**Results:**

-Does the analysis presented match the analysis plan?

-Are the results clearly and completely presented?

-Are the figures (Tables, Images) of sufficient quality for clarity?

Reviewer #1: Results are clearly presented and tables are appropriate.

Reviewer #2: In this study, the authors describe sampling of various bat species in southern Kenya, and report on the prevalence of coronavirus RNA sequences among the sampled population. In general, I think there is a great need for work like this, to explore the coronavirus diversity that may one day spill over to other species including humans.

That said, there are some problems here - I think all of them can be fixed, but it will take some work and will not require additional bat sampling. There is the kernel of a useful paper here, but it could be so much better with additional analysis, and the data need to be much more fully presented.

1. Please add genome maps, showing the parts of each new genome that were recovered. Ideally, this would be done to scale, and at the domain level rather than dividing the genome into overly broad regions like pp1a/pp1ab. New strains often show novelties in some of the accessory genes, and I don't recall a good study on domain-level conservation in the region of pp1a before nsp4 - providing a map of what you found will make this data more useful to the field. For example, I can't tell if you were able to assemble any complete or coding-complete genomes or just partial genomes, and I can't tell how successful your bridging PCR was at closing the gaps. Full or coding-complete genomes can potentially be classified as new species by the ICTV, but from what I understand, much of the material available in the literature is fragmentary to some extent. It would be nice to see your new viruses get the wider attention that I think they deserve.

2. Please make the data available and add a table showing the name and accession number for each new genome or fragment. There is potentially some interesting information here about virus transfer within and among species of bats in Kenya. I would very much like to see an attempt at this sort of analysis added to the manuscript.

3. Table 1 would be better with male/female stats added. It's fine to keep the reproductive status, but break that down by sex, especially since you appear to be modelling the effect of sex on positivity in table 3. The title seems to suggest sex data was intended to be in this table originally.

4. It is not clear what the reference data in Fig. 4 refers to. Data from previous work? Other studies? If there is a source for this data, please cite it in the table, and if this is new data, please describe where/when it was collected.

5. Is the miseq data from these experiments going to be publicly available, and if so, what are the accessions?

6. In figure 2, what sort of data is being used? Is this amino acid or nucleotide? How long is the aligned contig? What are the boundaries, i.e., does the RdRp here include the NiRAN domain? There is a rather large polytomy in the center, which makes me suspect the data are from a rather narrow region.

7. In figure 3, which parts of pp1a are being aligned, and are they homologous? Also please show your new viruses in a different color so they stand out. Same question for pp1b and S - are these parts, entire regions, do they cover important domains? Too many questions unanswered.

8. In figure 4, are these the only sequenced parts of the genome? The only compared parts? It isn't clear, and two of the parts seem to overlap - does that mean the same data was used twice? The list of confidence values needs a lot more explanation to be useful. Assuming there is a bit of error in the trees due to undersampling and sampling bias, I don't really think any of these trees are different enough in topology to be very conclusive.

9. There aren't scale bars on any trees - that makes interpretation difficult.

10. On the supplemental figures, it isn't clear how they relate to Figure 2, or the small trees in Fig. 4 for example. And please color your new viruses or mark them in some way so they can be spotted. Are only these 4 Taita viruses used because they are the most complete? Something like AliStat can be used to give nice statististics relating to the quality of the underlying alignments behind the trees, and there are a number of statistics including TreeDist that would give you a way to compare the similarity or lack thereof between the various trees presented here.

11. Is there evidence, perhaps from other studies, that suggests other types of coronavirus such as Betacoronavirus would have been detected if present? There are some very interesting sarbecoviruses reported from Kenya in other studies, for example.

12. A more full discussion of bat sampling in Kenya, Tanzania, Mozambique and throughout eastern Africa would be a welcome addition to the discussion section.

Reviewer #3: The results and figures are adequate and clear.

**Conclusions:**

-Are the conclusions supported by the data presented?

-Are the limitations of analysis clearly described?

-Do the authors discuss how these data can be helpful to advance our understanding of the topic under study?

-Is public health relevance addressed?

Reviewer #1: Yes, conclusions appear valid and authors discuss the need for additional surveillance and ultimately gaining an understanding of potential public health relevance.

Reviewer #2: See above - I would need a lot more information to properly judge the conclusions, but on the surface everything seems reasonable here.

Reviewer #3: The conclusions are supported by the data presented and review of the literature. There could be more information to put these into context, e.g. on the importance of phylogeny of these genomes and on specific gene sequences to their potential to emerge; on the ecology of the bat host species and therefore the likelihood of spillover; and on relationship to alpha-CoVs known to infect livestock.

**Editorial and Data Presentation Modifications?**

Reviewer #1: Minor revision

Reviewer #2: See Results

Reviewer #3: None needed

**Summary and General Comments:**

Reviewer #1: The manuscript presented by Ogola and colleagues described molecular surveillance for coronaviruses in bats collected from locations in southeastern Kenya. This is an important problem for study and manuscript is very well written and clearly presented. Similarly, the methods described for detection and analysis of coronavirus sequences appear appropriate and well executed. Overall, I consider this an important contribution to the literature on distribution of bat coronaviruses in Africa. I offer the following small suggestions:

It would be useful to describe the spatial relation between bat capture sites and human habitations. Table 2 lists “buildings”, but were these human dwellings, commercial buildings, hospitals or what? Similarly, please indicate how many geographically distinct sites were subjected to surveillance - a small map of locations in that county would be appropriate but not necessary.

Line 23: “Bat species-specific RNA prevalence was ..” – would this be more clear as “The prevalence of detecting coronavirus RNA for specific species of bats was ..”

Line 42: Suggest changing “Our bat alphacoronaviruses detected..” to “The bat alpha coronaviruses we detected..”

Line 124-6 describes the PCR primers used to detect CoV. The reference is almost 20 years old and, if appropriate, it would be good to add a comment that these primers are known to detect all 4 genera of coronaviruses currently known (that is, we’ve learned a lot about new coronaviruses over since 2007 and it would be confirm those primers will not miss detection of recently discovered coronaviruses)

Reviewer #2: See Results. There is the kernel of a very nice study here, but the presentation needs lots of work before it can be completely reviewed.

Reviewer #3: Major comments

Overall this is a solid paper reporting whole genomes from some interesting alpha-CoVs in Kenya. The discussion could be expanded and made more informative by reference to the following:

• Some information on the likely ability of these new CoVs to infect people (from genome data/phylogeny)

• Information on ecology of bat host species that might have relevance for spillover

• Reference to the ability of other alpha CoVs to infect livestock, recently (e.g. SADS-CoV) or historically (reviewed in various papers by Saif & most recently in PNAS Keusch et al.)

Minor comments

L 64 – should be ‘likely zoonotic’ – I think they probably are but not yet definitively proven

L69 – these are just some of the reasons for zoonotic spillover of CoVs – others are ecological/demographic – human interactions with bats, environmental changes like deforestation and the unsustainable wildlife trade.

PLOS authors have the option to publish the peer review history of their article (what does this mean?). If published, this will include your full peer review and any attached files.

Reviewer #1: No

Reviewer #2: No

Reviewer #3: No

**Figure resubmission:**

**Reproducibility:**



---

## [Decision Letter · Decision Letter 1]

23 Jul 2025

Detection and genetic characterization of alphacoronaviruses in co-roosting bat species, southeastern Kenya

Dear Dr. Ogola,

Thank you for submitting your manuscript to PLOS Neglected Tropical Diseases. After careful consideration, we feel that it has merit but does not fully meet PLOS Neglected Tropical Diseases's publication criteria as it currently stands. Therefore, we invite you to submit a revised version of the manuscript that addresses the points raised during the review process.

Please submit your revised manuscript within 60 days Aug 22 2025 11:59PM. If you will need more time than this to complete your revisions, please reply to this message or contact the journal office at plosntds@plos.org. Please include the following items when submitting your revised manuscript:

We look forward to receiving your revised manuscript.

Kind regards,

Michael W Gaunt, PhD

Academic Editor

Andrea Marzi

Section Editor

Shaden Kamhawi

co-Editor-in-Chief

Paul Brindley

co-Editor-in-Chief

**Additional Editor Comments:**

**# AE Editorial Note on Recombination Claims and Phylogenetic Analysis**

Both Reviewer 2 and I continue to have serious concerns regarding the authors’ phylogenetic analyses, which represent a demonstrable weakness in the manuscript. This directly affects the validity of the authors' claims, especially the assertion in the Abstract:

> "Incongruent clustering patterns based on distinct genomic regions suggest that this virus may have undergone recombination events during its evolution."

Despite previous feedback, the authors continue to assert recombination without sufficient supporting evidence. This claim must be removed unless a substantially stronger analytical foundation is provided.

1. Lack of Evidence for Recombination

The authors have relied on RDP software but appear to have used only a subset of available methods. They have excluded key approaches within RDP that account for stochastic noise, such as those that handle variation at the third codon position. The analysis is further weakened by reliance on incongruent trees over small genetic distances (few mutations) that are not convincing and a sliding window similarity analysis (Fig. 5b) without appropriate correction.

Specifically, the use of raw nucleotide "similarity" as a metric (described on the y-axis in Fig. 5b) is outdated and introduces known artefacts. This issue was addressed over 50 years ago by Jukes and Cantor (1969), whose correction is foundational in modern molecular phylogenetics. The authors’ approach in Fig. 5b disregards this standard and constitutes a major methodological flaw.

2. Authors’ Justifications – Inadequate and Misapplied

The authors’ defence of their recombination claim is based on three points:

a. Multi-clonality

The sequencing and assembly approach used assumes clonality, yet the authors now claim evidence of multi-clonal viral populations. However, no bioinformatic or biological methods were employed to remove/separate mixed infections or identify haplotypes. It is far more likely that the apparent signal of recombination stems from assembly artefacts caused by mixed reads than from biological recombination. Quality control metrics for the NGS data are also lacking.

b. Intra-species Sampling; see point 3

c. Coronaviruses are “known” to recombine

Line 291 of the manuscript states:

> “Since coronaviruses are known for their capacity for homologous recombination \[62]...”

While recombination has been documented in coronaviruses, the frequency of such events in natural populations remains low. For example, despite billions of SARS-CoV-2 infections, confirmed recombinants are limited. The biological capacity for recombination does not justify claiming its presence in the absence of compelling evidence in the specific dataset.

3. Intra-species Sampling

The authors argue that recombination is far more likely within a single species. However, this claim is both historically and methodologically weak. A relevant example is dengue virus: serotypes of dengue (now considered distinct species) produced strong signals of phylogenetic incongruence in early analyses, and these results were published in high-impact journals over two decades ago. These early findings even contributed to the development of recombination detection tools that rely on incongruent tree topologies.

Yet despite this historical precedent, the field has since shifted. Notable leading researchers — including Vasilakis and Weaver — now argue that dengue virus does not, in fact, recombine, or does so only rarely. This illustrates the danger of prematurely labeling incongruent signal as recombination: once such claims are in the literature, they are difficult to retract or revise, even as analytical tools and interpretations evolve. The authors' current data and methods fall below those early dengue virus studies, making their recombination claims concerning.

4. The Broader Implication

The current manuscript presents weak and methodologically outdated evidence to support a claim of recombination. If published, this would lower the analytical standards of the field and risk reinforcing inaccurate evolutionary narratives. Indeed, the star-like phylogeny in Fig. 2 might itself result from unresolved multi-clonal signals, further complicating interpretation.

5. Required Action

The authors must:

Remove all claims of recombination unless they are supported by robust phylogenetic and statistical evidence.

Deposit raw NGS reads and associated metadata in a public repository (e.g., NCBI SRA).

Provide appropriate quality control metrics and explain assembly parameters, particularly with respect to potential mixed infections.

This is the final opportunity to address these issues. If the current language and analyses remain unchanged, the manuscript risks rejection on grounds of methodological inadequacy and potential to mislead.

**Journal Requirements:**

1) Please upload all main figures as separate Figure files in .tif or .eps format. For more information about how to convert and format your figure files please see our guidelines: 

2) Please amend your detailed Financial Disclosure statement. This is published with the article. It must therefore be completed in full sentences and contain the exact wording you wish to be published.

2) If any authors received a salary from any of your funders, please state which authors and which funders..

3)  Please ensure that the funders and grant numbers match between the Financial Disclosure field and the Funding Information tab in your submission form. Note that the funders must be provided in the same order in both places as well.  

**Reviewers' Comments:**

Reviewer's Responses to Questions

**Key Review Criteria Required for Acceptance?**

**Methods**

-Are the objectives of the study clearly articulated with a clear testable hypothesis stated?

-Is the study design appropriate to address the stated objectives?

-Is the population clearly described and appropriate for the hypothesis being tested?

-Is the sample size sufficient to ensure adequate power to address the hypothesis being tested?

-Were correct statistical analysis used to support conclusions?

-Are there concerns about ethical or regulatory requirements being met?

Reviewer #1: The authors have done a good and appropriate job of responding to reviewer comments, adding clarity to several issues on phylogenetic and ecological analysis.

Reviewer #2: Open questions – how are the trees rooted? It seems like there may be different answers for Figs. 2, 4 and 5.

What were the criteria for showing polytomies as opposed to distinct branches – it seems like a rule of some sort was being applied, maybe by default, and possibly sensibly, but it would help me to see it articulated.

**Results**

-Does the analysis presented match the analysis plan?

-Are the results clearly and completely presented?

-Are the figures (Tables, Images) of sufficient quality for clarity?

Reviewer #1: Yes, analysis is well described.

Reviewer #2: Presentation is not particularly slick, but does what the authors and reader need, I think.

**Conclusions**

-Are the conclusions supported by the data presented?

-Are the limitations of analysis clearly described?

-Do the authors discuss how these data can be helpful to advance our understanding of the topic under study?

-Is public health relevance addressed?

Reviewer #1: Conclusions appear valid

Reviewer #2: The conclusions seem reasonable to me, with the possible exception of the recombination analysis.

**Editorial and Data Presentation Modifications?**

Reviewer #1: No issues noted

Reviewer #2: I’m not sure how robust or useful the recombination analysis is. I defer to the other reviewer’s expertise on that. If it was me, I’d probably delete panels C and D and the corresponding supplementary information. It’s just a matter of having very thin data for the whole-genome recombination analysis – these trees have 8 or fewer nodes, 5 or fewer if you factor in polytomies and near-identical sequences – and I don’t think there is enough signal there for a meaningful RF analysis of the resulting trees.

I also suspect BFR-C and BFR-F are picking up known hypervariable regions of nsp2 and nsp3, respectively, rather than genuine recombinant breaks.

**Summary and General Comments**

Reviewer #1: This manuscript has been significantly improved and provides a valuable contribution to the study of bat coronaviruses.

Reviewer #2: This is better than the last time I saw it. A quick blast against named coronavirus species shows that four most complete sequences (X193, X152, X162 and X167) all likely belong to the same species, which is distinct from previously described species, which adds to the value of the study. They do fit into a known subgenus, by the look of it, as the authors say, but it has a surprisingly broad global distribution, which I think is interesting.

The addition of what will likely be a new species is still the major selling point for me, as is the co-roosting species sampling design. The grouping of sequences by host species in Fig. 2 would seem to show more evidence that cross-species transmission is rare, with longer periods of host-specific adaptation between rare species jumps, despite some degree of co-roosting. Mostly, I like that you looked at this question.

PLOS authors have the option to publish the peer review history of their article (what does this mean?). If published, this will include your full peer review and any attached files.

Reviewer #1: No

Reviewer #2: No

**Figure resubmission:**

**Reproducibility:**



---

## [Decision Letter · Decision Letter 2]

18 Oct 2025

Dear Dr Ogola,

We are pleased to inform you that your manuscript 'Detection and genetic characterization of alphacoronaviruses in co-roosting bat species, southeastern Kenya' has been provisionally accepted for publication in PLOS Neglected Tropical Diseases.

Best regards,

Michael W Gaunt, PhD

Academic Editor

Abdallah Samy

Section Editor

Shaden Kamhawi

co-Editor-in-Chief

Paul Brindley

co-Editor-in-Chief

Reviewer's Responses to Questions

**Key Review Criteria Required for Acceptance?**

**Methods**

-Are the objectives of the study clearly articulated with a clear testable hypothesis stated?

-Is the study design appropriate to address the stated objectives?

-Is the population clearly described and appropriate for the hypothesis being tested?

-Is the sample size sufficient to ensure adequate power to address the hypothesis being tested?

-Were correct statistical analysis used to support conclusions?

-Are there concerns about ethical or regulatory requirements being met?

Reviewer #1: Now that the analysis for recombinational events has been removed, the methodology appears sound. There are no concerns relative to sample size or ethics.

Reviewer #2: I think the manuscript is much improved, with the biggest improvements being to the technical side of the genome analysis and phylogenetic analysis. I have a much clearer idea of what was done, and to me, the statistical analyses that were used seem appropriate (though I would defer to a more knowledgable reviewer in that respect).

**Results**

-Does the analysis presented match the analysis plan?

-Are the results clearly and completely presented?

-Are the figures (Tables, Images) of sufficient quality for clarity?

Reviewer #1: Yes, results now appear to be complete.

Reviewer #2: The results look reasonable. The supplemental figures probably belong as supplemental figures, but this being PLOS, they will be easily available to anyone who wants them.

**Conclusions**

-Are the conclusions supported by the data presented?

-Are the limitations of analysis clearly described?

-Do the authors discuss how these data can be helpful to advance our understanding of the topic under study?

-Is public health relevance addressed?

Reviewer #1: Yes, conclusions now appear justified.

Reviewer #2: After this revision, I think the conclusions are now fully supported by the results. It is still likely a new species, and should be brought to the ICTV coronavirus study group as such.

**Editorial and Data Presentation Modifications?**

Reviewer #1: No issues noted.

Reviewer #2: I like the new version.

**Summary and General Comments**

Reviewer #1: Now that the suspect recombination analysis has been removed, the manuscript appears to be a valuable contribution.

Reviewer #2: This looks ship-shape to me.

PLOS authors have the option to publish the peer review history of their article (what does this mean?). If published, this will include your full peer review and any attached files.

Reviewer #1: No

Reviewer #2: No

---

## [Editor Report · Acceptance letter]

Dear Dr Ogola,

We are delighted to inform you that your manuscript, "Detection and genetic characterization of alphacoronaviruses in co-roosting bat species, southeastern Kenya," has been formally accepted for publication in PLOS Neglected Tropical Diseases.

Best regards,

Shaden Kamhawi

co-Editor-in-Chief

Paul Brindley

co-Editor-in-Chief
